# Towards a Reasoned Glossary of Green Conservation: A Semantic Review of Green-Oriented Terms in the Field of Cultural Heritage

Davide Del Curto  and Anna Turrina *

Department of Architecture and Urban Studies, Polytechnic of Milan, 20133 Milan, Italy;
davide.delcurto@polimi.it
* Correspondence: anna.turrina@polimi.it

**Abstract:** The concept of green conservation has become a popular expression in parallel to the inflated development of innovative green practices and products for cultural heritage. However, the absence of a consistent definition of emerging green concepts does not prevent the dilution of the terms in scientific research and commercial propaganda. On this basis, this article discusses the several meanings of the concept of green conservation and its related terms toward a viable and comprehensive definition. The semantic analysis relies on the identification of different sources to evaluate the coherence of the meanings in usage in scientific and non-scientific domains. Search terms—such as "bio", "eco", and "green"—were collected and classified into lemmas of emerging constructs. Lastly, two glossaries resulting from the two main sources provide a comparative analysis to evaluate the degree of intersection and divergence among equal terms. The research drew from over 100 studies and five international databases to generate a hierarchical classification among 220 constructs and to identify six definitions of green conservation. This paper contributes to greater clarity and encourages a semantic discussion toward a common vision for a green conservation perspective for future research and informed preservation practice.

**Keywords:** green conservation; cultural heritage; terminology; sustainability; greenness; green materials; standardization; databases

## 1. Introduction

Over the last ten years, the concept of green conservation entered into common parlance to represent the development of innovative practices and products in the field of cultural heritage [1]. This trend can be seen in the rapid growth of scientific articles on this topic and in the spread of green-oriented terminology in the construction market. It embraces terms such as bio-based, eco-compatibility, greenness, eco-friendly, and others while involving the environmental dimension of sustainability in historic preservation practice. Rarely has a concept gained status as rapidly as the term green in such many fields. However, the novel joining together of "green" and "conservation" is not without its problems. Where "green" is often understood as a commercial opportunity and a synonym for "ecological". In fact, the absence of an acknowledged and consistent definition does not prevent the abuse and dilution of the term in scientific research and commercial propaganda. This lack of clarity also applies to a high number of terms that are interchangeable without coherence, such as green products and sustainable materials. This semantic blurriness leads to imprecise definitions and improper usage. Another important issue is that most of the words are multiterm units and therefore, the definitions are unavailable in dictionaries. However, where green conservation is conceived in its comprehensive sense as an unfolding of potential, there it serves to add dynamism to the concept of sustainability in cultural heritage.

On this basis, this article provides the results of a semantic survey and summarizes the different definitions of the concept of green conservation, and its implied keywords, to reconstruct a verifiable and comprehensive meaning of the term. This study can be seen as an attempt to answer the following questions: What do you understand by the expression "green conservation"? Is it clear or nebulous? Too vocal"? A "catch-all term"? All these open issues are at the core of sustainability and the built heritage field at this moment when green conservation initiatives are trying to trace a new contour in the field. Though this concept seems to be acknowledged in the green chemistry discourse (i.e., elimination of toxic compounds, the recycling factor, etc.) and traceable to common practices in heritage conservation, it still represents one of the most significant translational problems of our time.

The paper's introduction offers a brief overview of the development of a green sensibility linked to the idea of the environment from a historical perspective to explain when the notion of green conservation was firstly conceived. It also serves to clarify the absence of previous reviews on the concept of green conservation. Reasons for this uncertain state of the art include its multidisciplinary nature and recent attention to the topic, as well as the impossibility of standardizing the case-by-case techniques and turning them into a single guideline [2].

The development of a green sensibility in the field of cultural heritage translating into green conservation means fully understanding the historic perspective in which the green dimension was born. This section prefaces the semantic review with a brief overview of the most significant milestones that have led to the origin of the notion of green conservation. At the same time, it clarifies the state of the art to underline why a reasoned glossary is needed and why it differs from existing semantic reviews. "The world is, fortunately, beginning to turn green…at least pastel green" [3]. With this phrase, American biologist Edward O. Wilson, also known for coining the term biodiversity in 1985, described in 2011 a trend that pervades modern society and introduces the steps by which the color green came into everyday usage. However, before attaining the contents of the modern concept, the term green was limited in its most literal sense to describing a specific disciplinary field, that of the environment and natural systems. Hence, the rise of what Grober defines as a green (or green-ish) popular culture [4] that manifests itself through environmental, economic, social, political, and cultural actions is a fairly recent phenomenon. For this reason, it should be stressed that when conservation was associated with the term sustainable development in 1980, it addressed only the protection of environmental resources and, specifically, it aimed to ensure the planet's capacity to "sustain the development" consisting of (a) the maintenance of ecological processes, (b) the preservation of genetic diversity, and, finally, (c) sustainable use of species and ecosystems [5]. With the advent of the green connotation, however, it seems that the spread of this term is almost taking on the same speed as the term sustainable.

The first historical introduction of the term green in everyday usage corresponds to the rise of global environmentalism in the second half of the twentieth century, claiming for cleaner and safer living conditions [6] and, more specifically, with the rise of green chemistry. The environmental issue, certainly, had already been addressed many centuries before within the sylviculture textbooks for responsible forestry management [7,8], but a formula for environmental protection did not yet exist. Officially, it is with the advent of the XX century that the idea of the environment was brought to light. In addition, it takes place in parallel to its loss. The air, so far forgotten by the industrial revolution and the chemical war, turns into a resource worth protecting [9]. From the agricultural to the chemical fields, the design of the non-objective experimented toward the improvement of the air quality. This historical moment, dotted with political agendas and non-governmental initiatives, for the first time focuses attention on the ecological condition of human existence. According to most of the storytelling on the sustainability movement [10–12], the 1960s and early 1970s seem to mark the rise of modern environmentalism. Bestselling books and global agenda initiatives are the most quoted results of this intellectual moment. The notion of green has

been consolidated by the publications of Carson's cult-defined book [13] and Anastas [14], the pioneer of Green Chemistry. Both contributions are the result of an American protest against the use of chemicals and the best-known popular and literary translation of the American narrative criticizing city pollution and environmental risk to humans and nature. In the first case, the term green, invoked by the American biologist, refers to an idyllic and romantic idea of the planet—"[...] green fields. A picture of the safe and secure word; but then comes the shock". In the second textbook, it becomes representative of a new disciplinary field, that of Green Chemistry, which in 1990 stands against traditional chemistry. The modern environmental movement finds further green connotations when the theme of protecting the planet reaches a global type of communication, and in the collective imagination among the "Earthy color tones", green is the one that basically recalls the natural dimension. The construction of this common imagery associated with green imaginings in the literature sees the appearance of the Green Revolution underway [15]. The literature on the topic, however, would indicate a spectrum of greens rather than a strict dichotomy: the ideology of all those shades along the spectrum of greenness is determined by their attitude to the environment. Green politics explicitly seeks to de-center the human being [16], question mechanistic science, and its technological consequences, to refuse to believe that the world was made for human beings.

Starting from this historical framework, the green conservation expression turns into a voguish concept in parallel to an ecological vision of cultural heritage. The conservation field was quick to claim that old buildings are innately green because of their embodied energy and climatically proper materials and structures. However, research proving this position has been slow in coming [17]. The green factor acquires momentum first in the new construction sector along with the rise of a low-carbon society [18] and, only afterward, has landed into the discourse of historic buildings [19]. Green as a recent parameter for cultural heritage [20] results in a lack of theoretical discussion and semantic review. At present, the terminology of evolving green design [21] is built on criteria that are not akin or properly extended to embrace all the restoration sector (i.e., LCA, recyclability, regenerative design, green cleaning). The language of green buildings still covers most of the existing vocabularies. Moreover, different materials platforms have been developed, such as the GreenSpec Directory [22] and the Oikos Green Building Source [23], or the Building Design Guide [24] that lists the green attributes of environmentally preferable products as follows [25]:

1. They promote good indoor air quality (typically reducing VOC emissions).
2. They are durable and have low maintenance requirements.
3. They incorporate recycled content.
4. They have been salvaged for reuse.
5. They are made of natural and/or renewable resources.
6. They do not contain CFCs, HCFCs, or other ozone-depleting substances.
7. They do not contain highly toxic compounds.
8. They are obtained from local resources and manufacturers.
9. They can be easily reused.
10. They can be readily recycled.
11. They are biodegradable.

All these features have been published in 2007, one year after the introduction of the European legislation REACH [26], Registration, Evaluation, and Authorization of chemicals, which established procedures for the collection and evaluation of information on the properties of substances and the hazards arising from them to support alternative green solutions. As a matter of relevant historical window, it is necessary to mention that the attributes above refer back to the 12 principles of Green Chemistry [14], which similarly involve all product stages.

However, fifteen years later, how fully understood is the concept of green conservation? Documented research exists on the semantic review of sustainability [27,28], or circular economy terms [29] and their definitions but, to our knowledge, the topic of green

conservation has not been analyzed yet. Although certain terms on the environmental value of restoration have been listed in a quite recent book [30], a systematic study has never been undertaken. The 2023 edition of the Italian pricelist of restoration works for cultural heritage [31] is the most visible example of this: despite the ample use of biotechnologies and nanotechnologies in the sector, none of the published entries include novel green products. In this view, it is hard to understand how to position an essential oil-based product, having the function of disinfectant, or an enzyme-based product for stone cleaning under the heading of biocide.

On this basis, the research unfolds across three more sections. Section 2 outlines methods and sources adopted to develop a twofold glossary, stretching from scientific to non-scientific sources. Within the analysis, both green-oriented constructs and explicit assumptions have been collected and classified to create a final revised version. Section 3 presents and discusses the results of the semantic analysis giving prominence to most recurrent terms, such as bio-based, eco-friendly, and green, classified into lemmas of emerging constructs. To obtain a general sense out of the glossaries, a comparative analysis has been carried out to evaluate the degree of intersection and divergence among equal terms. Lastly, Section 4 is dedicated to the discussion of the limits of the study and concludes with future research advances. The following work can contribute to greater clarity and encourages a semantic discussion toward a common vision from a green conservation perspective.

## 2. Materials and Methods

The study of this paper is anchored in the interpretative research paradigm, based on exploratory inquiry, that is conducted through the collection and selection of relevant terminology in the view of a final evaluation of the different meanings of green conservation. Following the model of semantic analysis, two investigations have been carried out in parallel to address both the scientific and non-scientific literature. Considering the diverse nature of each source, our way to approaching the next subsections has been proposed in alignment with the approach adopted for each of the two domains. This section is divided into two sub-sections. First, we describe how we gathered 77 definitions retrieved from the scientific literature on green conservation. Second, we explain how we collected 51 definitions published in non-scientific literature. The sample size has been deemed sufficient to establish a correct comparison in the next phase.

### 2.1. A Glossary from Scientific Literature

The first source of the analysis, namely the scientific literature, consists mostly of journal articles where preservation or green chemistry experts are the perceived audience and, in a minor part, of books. Many scientific studies related to green conservation have been published in journals rooted in the discipline of material science, chemistry, and green conservation, prompted by a quite recent forum on green strategies for heritage preservation, the so-called Green Conservation Conference. With the intention to develop a representative sample of green conservation definitions, the first action is based on the analysis of the proceedings of the Green Conservation Conference for Cultural Heritage. Since this forum is still the only explicit academic player to dedicate to this topic, much of the work of this glossary is driven by the research presented during the congress and later published in peer-reviewed articles.

To better understand the reasons for this bibliographic selection and its inclusion criteria, an overview of the conference needs to be mentioned. The first initiatives promoted on the topic of green and sustainable conservation began in Rome through a series of conferences, initially presented as "Which sustainability for restoration?", "Sustainable restoration 2.0". In 2005 the final form was titled Green Conservation Conference of Cultural Heritage to gather national and international experiences around the research of alternative products and technologies. The forum, now in its fifth edition in 2023, was founded by YOCOCU, YOuth in COnservation of CUltural Heritage, an association of

professionals and researchers in the field of cultural heritage conservation based in Rome who focuses on innovations in green chemistry, and not only, applied to conservation. The association's mandate is to promote and consolidate an increasingly active network of green and sustainable research related to cultural heritage through the organization of international seminars. The success of its activity can be seen in its ability to expand the initial perimeter of the conference toward a broader vision of the meaning of green. Over the years, we have witnessed an evident evolution of the topics toward a new consistent idea of green conservation in the heritage field. To detect this change, our approach involves the definition of a comprehensive view of the research conducted so far by scientific experts. Because there is no uniform promotion of the forum by the association nor a complete overview of its structure, the different topics have been collected in a unique sheet and analyzed in terms of overlapping arguments. Information reported has been extracted both from the editorials of special issues and the conference programs: additionally, the number of papers is specified for each session. The following diagram (Figure 1) provides the number of papers for each topic/session along the four editions of the conference 2015–2022. Given the many-faceted titles of the sessions, a second subdivision was deemed necessary to differentiate biotechnologies (green) and nanotechnologies (blue) articles from research covering not only the environmental but also social, economic, and cultural values. This process serves to explain the evolution of items presented during the conference under the cluster of cultural heritage and to measure the focus on built heritage research. It further clarifies the direction of interests enclosed in the concept of green conservation: while previously, the first two editions presented the applications of green chemistry on cultural heritage, now the concept expands further to embrace sustainable design and practices. Put another way, the green conservation conference started including both environmental and cultural sustainability.

Once understood in the dimension of this literature sample, a systematic assessment of definitions was developed to collect the terms and to verify the coherence of their usage. To address this, we considered not only explicit definitions of green conservation along the text of scientific papers but also the neighboring green terms. In this way, the collection included a broader spectrum of meanings to support one consistent common definition within a single article. In order the facilitate the reconstruction of this common view, a glossary was adopted to collect the data and to facilitate the comparison of the terms toward the identification of common constructs. Different types of narration have been considered, both implicit and explicit to the term. In short, the methodology (Figure 2) tracks the following actions:

1.  The identification of the research context among the topics of the conference to chart the evolution of green conservation storytelling over the years.
2.  The definition of three main categories for recurrent topics to merge the papers under the same session and avoid overlapping.
3.  The construction of the glossary through a series of different entries that consider (a) the lemma to which the term belongs, (b) the conference year, (c) the bibliographic reference or authors, (d) the scientific field among the three previous categories, (e) the term and, where applicable, the object of the construct, and lastly, (f) the as-found definition.
4.  The completion of the semantic collection, with an accent on the difference between explicit and implicit descriptions for each term.
5.  The identification of definition synthesis.

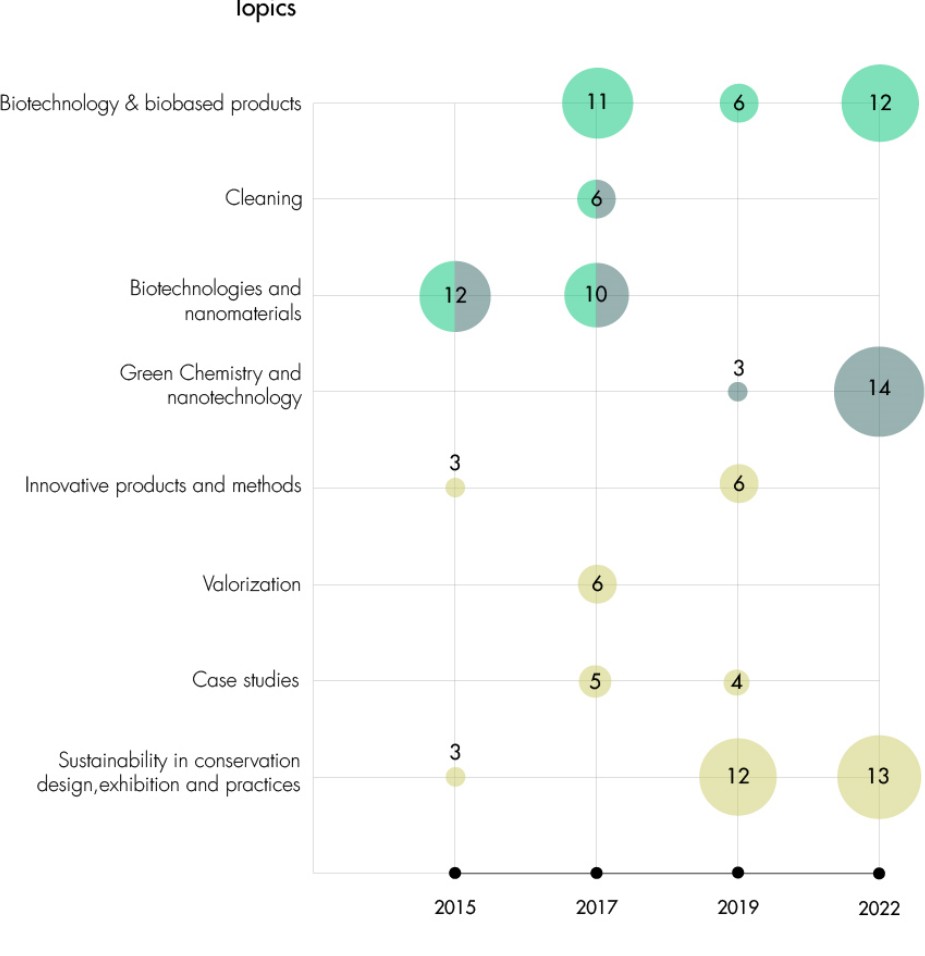

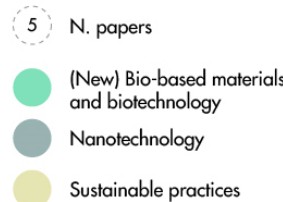

**Figure 1.** Topics comparison over the four editions of the Green Conservation Conference 2015–2022. The table reports the number of papers for each topic, and it combines overlapping sessions under three main categories: new bio-based materials and biotechnology, nanotechnologies, and sustainable practices.

This classification framework is not conceived to be only a system for collecting green-oriented terminology. Rather, the glossary entries should support the semantic analysis in parallel to the historical and subjective evolution of the concepts. In this way, the definition itself can be evaluated according to the authors' field, the cultural heritage object, and the year of the study.

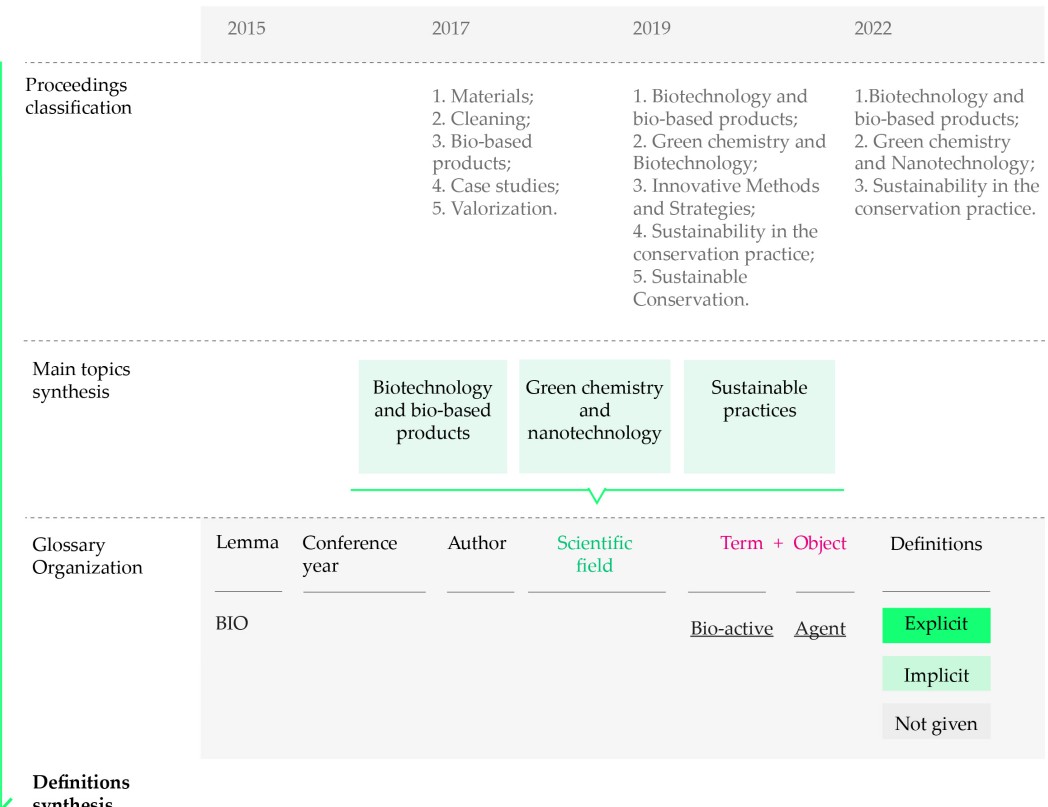

**Figure 2.** Road map of the semantic survey for the collection of terms from the Green Conservation Conference for Cultural Heritage proceedings.

### 2.2. A Glossary from Non-Scientific Literature

On the other side, the non-scientific literature represents a non-secondary source for the research. In this section, the analysis focuses on the terminology developed by governmental and non-governmental bodies, namely the International and European Agencies for Standardization [32]. The use of innovative technologies and new green materials and products is indeed growing. The increasing demand for full-scale commercialization has been met with many uncertainties, and the market needs to consolidate the state-of-the-art to enable commercialization and a wider-scale application of novel products and technologies. For this reason, the action of standardization institutes and agencies play a crucial role in the development of standards and tools to provide guidance to the business sector and research. Official and clear definitions of terms can be found in the form of databases or vocabularies—available on web platforms in most of the cases—mainly conceived for product declaration and substance risk management.

In order of year of foundation, the following five organizations have been selected:

1. ISO, International Organization for Standardization (1947) [33] is a worldwide federation of national standards bodies (ISO member bodies) representing 167 countries. International Standards cover many areas of technology, management, and manufacturing and are available in a terminological browsing database, the ISO Online browsing platform [34], and Electropedia [35], a second platform developed in collaboration with IEC, the International Electrotechnical Commission.

2. CEN, European Committee for Standardization (1961) [36] gathers the National Standardization Bodies of 34 European countries in relation to a wide range of sectors. In the specific field of materials, the association can be considered the main reference when dealing with the concept of a bio-based product. Due to the lack of standards for many novel products, the European Commission has mandated CEN to conduct new standardization work in the area of bio-based products. Recently, in addition to gen-

eral horizontal standards for bio-based products, CEN has been developing standards for specific fields of application, such as bio-polymers and bio-solvents. CEN/TC 411 is the Technical Committee responsible for conducting new standardization work in the area of bio-based products, and covers many aspects, such as (a) terminology and communication and (b) standardization of sustainability criteria. In this case, a specific standard for bio-based products was published in August 2014 [37].

3. EPA, United States Environmental Protection Agency (1970) [38] works to ensure that chemicals in the marketplace are reviewed for the safety of human health and the environment. Along with research activities, the agency provides a public database coming in the form of a Science Inventory [39] of products.

4. EEA, European Environment Agency (1990) [40] coordinates environmental information and integrates environmental considerations into policies moving toward sustainability. Since 2017, it has provided a list of environmental terms on a searchable web database and on a published glossary of Bioeconomy within a report on bio-based products [41].

5. ECHA, European Chemicals Agency (2007) [42], is a central agency that works on the sustainable management of European chemicals following the EU regulation called REACH (EC 1907/2006) [26]. In doing so, the agency hosts one of the largest public databases on chemicals in the world (more than 245,000 chemicals) to serve informed choices of companies, researchers, industries, and consumers. The multilingual terminology in the chemicals field is offered in all official EU languages to ensure compliance with obligations under EU chemicals legislation. IATE [43] for Interactive Terminology for Europe is the central EU terminology database used by the language services of ten institutions and bodies and managed by the Translation Centre. The platform was launched in 1999 (finalized in 2004) and fully rebuilt in 2018 to provide a web-based infrastructure for all EU terminology resources. From the point of view of usability, this dynamic inventory is unique for providing the term definition combined with the synonyms, context, comments, definition source publication, publication year, and main subject (Table 1).

**Table 1.** An overview of the IATE inventory released by ECHA: the term eco-sustainable.

| Term: Eco-Sustainable | Domain: Environmental Policy |
| --- | --- |
| **Entries** | |
| IATE ID | 3577083 |
| Owner | EESC/COR |
| Definition | using methods, systems, and materials that minimize impact on ecosystems |
| Definition reference | COR/EESC-EN [1] |
| Context reference | Opinion of the EC of the Regions [2] |
| Reliability level | ★★★ [3] |

[1] [44]. [2] [45]. [3] The reliability value indicates the consistency of terminology in EU documents referring to the reliability of the sources used. Value goes from one-star reliability (★ reliability not verified) to four-star reliability (★★★★ very reliable) when terms are well-established and widely accepted by experts as the correct designation, or confirmed by a trusted and authoritative source, in particular a reliable written source.

As for the collection of terms, the same approach of previous scientific proceedings has been adopted; therefore, a second glossary of non-scientific literature was conceived starting from the terms extraction from standards and glossaries of the selected bodies. A datasheet has been arranged to host (a) the term and its object, (b) the year of the publication, (c) the agency from which the term has been extracted, (d) the name of the digital database and (d) the as found definition.

The two glossaries come into a datasheet format to allow the next phase of the work that attempts to identify clear definitions out of the heterogeneous sources and the remarkable number of terms found in both literatures. As stated, many glossaries on the meaning of sustainability or green market still exist. However, the novelty of this process relies

on the perimeter of the research that focuses on cultural heritage, yet some of the sources belong to other fields.

## 3. Results

As for the first glossary coming from the analysis of scientific literature, a total of around 120 articles have been selected, and eight main lemmas have been identified to represent the basic form of the terms linked to the concept of green conservation. The key constructs survey represents the preliminary action before the synthesis of definitions. Key constructs or green-oriented terms can be seen as the first linguistic tool to define the notion of green conservation. By reviewing the green terms provided by the authors of the Green Conservation conference between 2015 and 2022, a total of eight recurrent lemmas have been found in different percentages. The most recurrent ones are "Bio" (33%), followed by "Green" (19%), "Eco", and "Sustainable" (14%) (Figure 3). Less used terms such as natural, recyclable, reusable, or preventive amount to 6–7%.

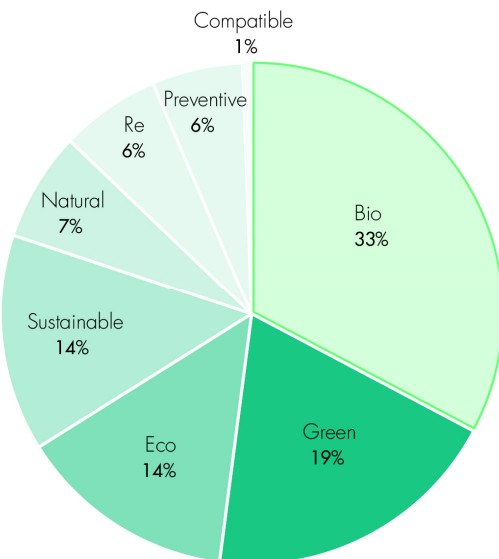

**Figure 3.** Percentage of uses of each lemma (prefix, roots, etc.) in the proceedings of the four editions of the Green Conservation Conference 2015–2022.

As can be seen from the diagram above, there is ample usage of terms within the conference proceedings, but these percentages drastically change if we consider the percentage of explicit definitions. In fact, if we further investigate the eight lemmas (in the form of prefixes or adjectives), we can detect evident variations (Figure 4). The use of the terms without any reference or description indeed affects all the constructs: it is no coincidence that the lemma of "Bio" includes a lot of constructs without clear definitions (56 uses in total, of which only 25 have been unfolded and explained in the body of the publication); on the other hand, those authors that rely on the use of "Sustainable"—24 papers in total, of which 16 with definition—and "Green"—33 papers in total, of which 22 with definition—seems to be more transparent in clarifying their intention. Looking into detail, in the specific case of bio-oriented terms (Figure 5), 21 different terms are in use, but some of them lack definition (i.e., bio-origin, biology-based, bio-mortar).

As for the non-scientific contributions, the analysis identified a total of 51 definitions, of which:

- Twenty-five definitions related to the lemma of Bio.
- Twelve terms related to the lemma of Bio (Bio-based, Bio-compatible, Bio-composites, Bio-degradability, Bio-degradable, Bio-degradation, Bio-economy, Bio-mass, Bio-material, Bio-mineralization, Bio-polymer, Bio-technology).
- Seven definitions of Bio-based from 2014 to 2020.

- Six terms related to the lemma of Eco (Eco-effectiveness, Eco-efficiency, Eco-friendly, Eco-logical, Eco-sustainable, Eco-toxicological).
- Four terms related to the prefix of Re (Recyclable, Recyclability, Reusability, Renewable).
- Three terms related to the lemma of Green (Green chemistry, Green-labeled product, Green material).

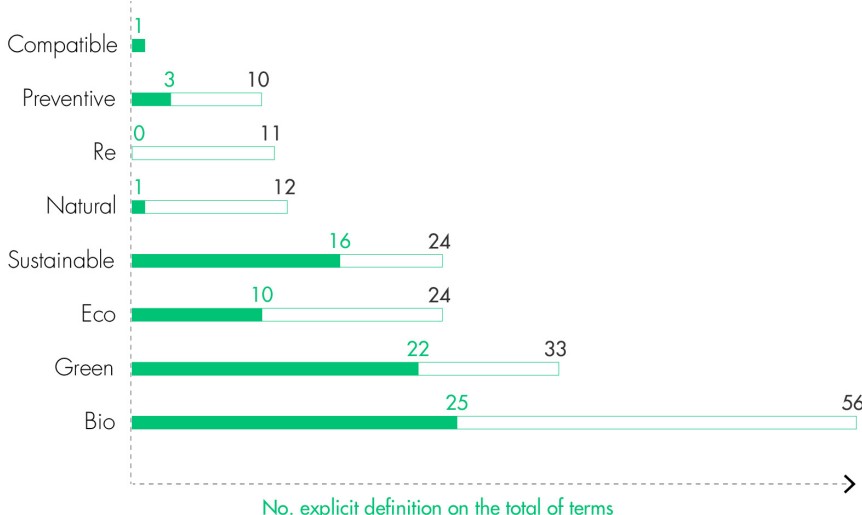

**Figure 4.** Number of uses of each lemma (in black) in the proceedings of the four editions of the Green Conservation Conference 2015–2022 and the number of explicit definitions (in green) reported for each lemma.

Main considerations of the results of the analysis concern the consistency in the meanings of the terms. In particular:

- CEN and EPA differ from EEA in describing the meaning of bio-based material or product. They adopt a clear distinction between bio-based and novel bio-based substances, recalling the long definite historical origin of the term.
- Insufficient definitions have been found in regard to the concept of green products or green material; while the green conservation terminology from scientific literature expresses itself without limits of neologisms, the agency's attitude strictly focuses on the safe path of bio-based technical codification.
- Many definitions found in the Inventory or Digital Vocabularies of agencies present references from scientific literature (i.e., bio-based definition from EPA's Science Inventory results from the Encyclopedia of Chemical Technology, or a second example is provided by the eco-effectiveness description from the IATE platform, which refers to a scientific paper on newest concepts).

Looking collectively at the terminology identified from scientific and non-scientific literature, this article concludes with a few brief remarks on the meaning of equal terms. The image below (Table 2) summarizes the number of different terms collected from the analysis considering the number of (1) words in total, the number of (2) words complemented with explicit definitions, and, lastly, (3) the number of explicit definitions for each lemma. While for the lemma of "Bio", the number of terms provided by the two glossaries corresponds to 25 definitions from both parts, less attention is given to other key constructs such as those under the lemma of green. Therefore, the comparative analysis has to face the limits of the survey due to the absence of a consistent body of literature on vocabulary for green and sustainable heritage and the absence of contributions from green marketing.

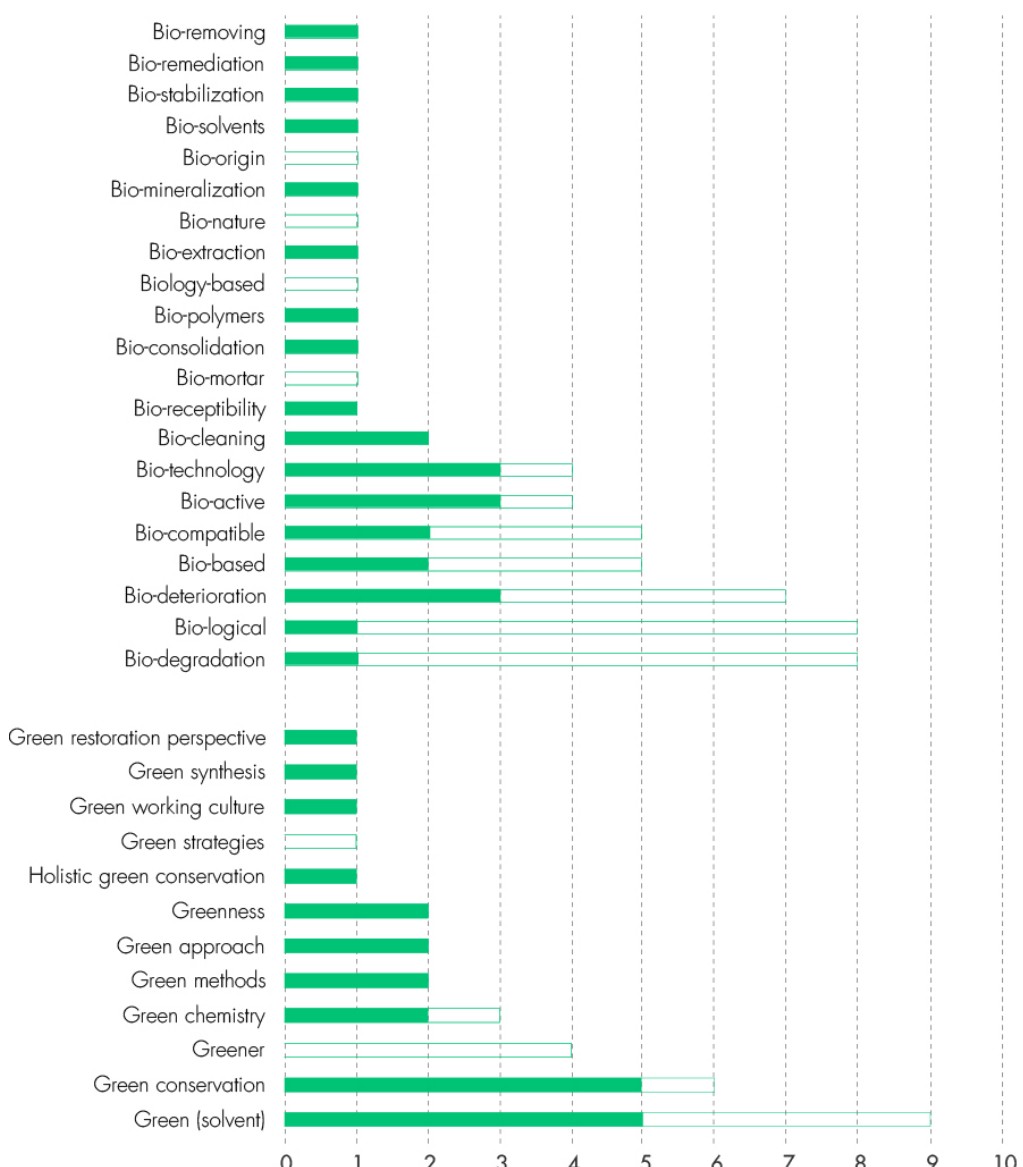

**Figure 5.** Recurrence of terms for each lemma and number of explicit definitions (in green) reported for each lemma.

**Table 2.** Overview of the terms collected from the two semantic analyses. A first quantitative comparison serves to underline the equal use of bio-oriented terms from both literature domains.

|  | Scientific Literature | Non-Scientific Literature |
|---|---|---|
| Different terms collected | 58 | 44 |
| Definitions in total | 77 | 51 |
| Bio definitions | 25 | 25 |
| Bio terms | 22 | 18 |

To reclaim a genuine version of the terms, a selection of the most significant and recurrent words from scientific literature and non-scientific literature has been made to evaluate the level of intersection between equal constructs.

### 3.1. Bio-Based Terms

The huge number of uses of the term claims a clarification toward a uniform and viable definition. In this regard, scientific literature defines a bio-based product as a substance that responds to sustainable requirements in terms of:

- The absence of interactions during the treatment, namely, it does not affect the porosity or existing fractures [46].
- Safety for humans and the environment [46].
- Optimization of natural microbial processes [47].

On the other side, the glossary of governmental and non-governmental agencies results in a more extensive description; firstly, they remark on the significant distinction between novel and pre-modern bio-based: firstly, CEN Technical Committee 411 defines bio-based products as "products from forestry and agriculture have a long history of application, such as paper, board, and various chemicals and materials [...] the last decade have seen the emergence of new bio-based products in the market" [36]; secondly, the EPA's Science Glossary adds its 2020 definition of "Many common materials, such as paper, woods [...] refer to bio-based materials, but typically the term refers to modern materials that have undergone more extensive processing" [39].

### 3.2. Green-Oriented Terms

When dealing with the lemma of green and related key constructs, the definitions have several facets. In synthesis, both the scientific and non-scientific literature identifies a set of properties defining a green product or material:

- Sustainable product [43].
- Minimal environmental impact and no toxic emission [43,46–52].
- Minimal impact on the operator's health [43,46–53].
- Involving the whole life cycle, minimal waste, maximum energy efficiency [43].
- Involving no organic solvents and promoting preventive conservation [48].
- Stability over time, reversibility, compatibility, minimally invasive [54].
- Affordability and availability at a larger scale, biodegradability, renewable origin, and recyclability [52].

### 3.3. Eco-Friendly

Another overused term, often lacking in definition, has been considered. Scientific literature is indeed sparse in this regard, and it seems very general: "[is] a green alternative to replace hazardous substances aiming at promoting sustainable development" [53–55]. The term sustainable development is called into question with no specific reference to heritage preservation practice; moreover, the definition merely means that eco-friendly simply deals with human and environmental safety and low toxicity. Similar content was provided by ECHA, the European Chemical Agency, in 2020. Among other institutions, this European body is the only one that offers within its interactive database a description of the concept, informing consumers about the use of eco-labels for green buildings. However, again, the meaning is reduced to a question of minimal harm to the environment and resource efficiency. No single standard yet exists for this term, and eco-labels are, therefore, too vague to be meaningful.

### 3.4. Green Conservation

Six published knowledge of green conservation were identified and consolidated through the contribution of many authors (Figure 6). Balliana and colleagues [48] stand out for their comprehensive treatment through products for cleaning, consolidating, and protecting surfaces. The article emphasizes the importance of "incorporating" green elements within architectural restoration practices and a focus on the entire product life cycle for holistic green conservation. Thus, not only the environmental issue but also the involvement of the economic and social aspects comes into play. This idea of green conservation

as a process is declined by Yoshida [56] in a different way. The author includes more than one definition, including the preventive factor in conservation; not only, but also economic and ecological aspects turn green conservation into a change of culture, or more properly, a lifestyle, that cannot be simplified to a technological, financial, or political approach. This sustainable way of working as a matter of communication is very far from the most recurrent definition of green conservation that focuses on the idea of green products. In order to bring more clarity to this concept, five sub-constructs have been identified to more detail the general definition of green material (definition n.4):

- (4.1) Stable, reversible, compatible [51–57].
- (4.2) No toxicity for operators and the environment [46–58], namely, the most popular construct ever. Several definitions suggest that the core of green conservation involves the safety of the operator and environment, providing different shades of the terms around the relationship of human nature. The introduction of the term "Greenness" in the latest publications emphasizes the role of "full safety" as a core goal of green conservation and is widely used to identify a diverse shade of sustainable cleaning. While these methods based their sustainability on a gradual release of the substance, green methods look for high boiling point products and the absence of hazard symbols. "greener" materials for cleaning procedures are supposed to accomplish the Green Chemistry principles, balancing minimum intervention and high performance at the same time [47–61].
- (4.3) Product safety applied to all the components employed (i.e., polymer matrix, solvents, and additional constituents), and all the materials should be recycled and degradable [1].
- (4.4) Recyclable and biodegradable [62].
- (4.5) Affordable and economically equal [49,63,64].

| Authors | 1 Process | 2 Working culture | 3 Prevention | 4 Product | | | | | 5 Environmental resources | 6 Historical techniques revival |
|---|---|---|---|---|---|---|---|---|---|---|
| | | | | 4.1 stable, reversible, compatible | 4.2 non toxic for operators and environment | 4.3 safety applied to all the component | 4.4 recyclable and biodegrdable | 4.5 affordable, economically equal | | |
| **Explicit** | | | | | | | | | | |
| Balliana et al., 2015 | X | | | X | X | | | X | | |
| Yoshida et al., 2015 | | X | X | | | | | | | |
| Ginè et al., 2019 | | | | | X | | | | | |
| Baudone, 2019 | | | | X | | | | | | |
| Lo Schiavo et al., 2020 | | | | | | X | | | X | |
| Passaretti et al., 2021 | | | | | X | X | X | X | | |
| Biribicchi et al., 2022 | | | | X | X | | | X | | |
| Sparacello et al., 2022 | | | | | X | | | | | |
| **Implicit** | | | | | | | | | | |
| Marin et al., 2016 | | | | | X | | | | | |
| Silva et al., 2016 | | | | | X | | | | | |
| Salvini et al., 2016 | | | | X | X | | | | | |
| Sgobbi et al., 2016 | | | | | X | | X | | | |
| Petrella et al., 2016 | | | | | X | | | X | | |
| Karadag et al., 2016 | | | | | X | | | | X | |
| Scheeper et al., 2019 | | | | X | | | | | | |
| Pasquale et al., 2019 | | | | | | | | X | | |
| Rubio et al., 2019 | | | | X | X | | | | | X |
| Alisi et al., 2021 | | | | | | | | | | X |
| Ganesan et al., 2022 | | | | X | X | | | | | |
| Strangis et al., 2022 | | | X | | X | X | X | | | |
| Macchia et al., 2022 | | | | | X | | | | | |

**Figure 6.** Summary of definition references abstracted to green conservation constructs [1,46–48,51–67].

Another definition involves a conscious use of renewable resources causing minimum environmental pollution and has a low-risk factor in relation to human health [65]; therefore, to accomplish the "green conservation criteria", the intervention should be specifically customized, also taking into account several environmental parameters such as light

exposure, humidity, quality of the air and so on. If all the five identified constructs are in some way related to environmental values, the cultural dimension is included in the green conservation methodologies as a "historical revival of traditional techniques" [66,67]. In this sixth abstracted construct, the term green founds its balance between technology and tradition. The advantage of using traditional techniques is the presence of non-toxic natural derivate and, consequently, compatibility and low interference with further conservative actions. All these six constructs are distinct yet interrelated.

## 4. Discussion and Conclusions

### 4.1. Limits of the Survey

This review offers an overview of the most recent research activities on counteracting green conservation practices, underlining all those aspects regarding compliance with sustainable criteria. The investigation of the definition of Green Conservation in the field of cultural heritage classifies the key constructs arising from the most common conservative procedures as well as guidelines for the development of innovative technologies. However, the risks or difficulties are twofold: on the one hand, the attempt to describe green conservation places this concept in a multiplicity of economic, social, and cultural dimensions (the three well-known pillars of sustainability) that make it difficult to be synthesized in one shape; on the other hand, there is a tendency to reduce this notion to a chemical approach. In this vision, a few limitations of this study should be noted:

- Uncertain testing: the difficulty of specific expertise to apply innovative products and still uncertain testing to verify the efficacy of products and reach common guidelines (e.g., in the case of essential oils for biodeteriogen removal, the various experiences are still not properly aligned).
- Placement of research: there still appears to be little dissemination and sectorial of research, so the topic is placed within the scientific journals of chemistry or other fields.
- Absence of a criterion for promoting outcomes: another element that underscores the absence of a homogeneous repertoire is the criterion by which experiences are recounted. These are often single-issue articles promoting the results of a single product; in other cases, there is an entire category united by a common function (e.g., biocides, adhesives) or chemical compounds.

Presence of non-exhaustive macro-groups: in the filing of research, especially with regard to scientific articles, two criteria for collecting results were found, as mentioned earlier, to be non-exhaustive and not covering all areas of research. For example, the first approach found is limited analysis of products by function, cleaning, consolidation, and protection, neglecting not only technologies but also process and design experiences that green conservation embraces. Certainly, it can be argued that the selection of contributions can be wider. The article is not providing a full and final definition of the notion of green conservation, but the set of definitions is at least fairly representative of the written definitions by academic and standardization bodies, given the approach adopted. It is, moreover, a possible input for further research and to direct attention to the critical use of the terms. Inclusion criteria varied within each knowledge synthesis, and therefore, newer articles are not likely to have been identified by the proposed method. The experiences of the green conservation conference have been selected in the semantic analysis for chronological reasons and their unicity. The research in semantic terms thus remains open, and the path toward the historical reconstruction of this Anglophone usage is rather uncertain.

### 4.2. Relation between the Terms

Starting from the fact that sustainability and green are not synonyms, despite wide abuse and interchange, it is clear that they are somewhat interlocked. The analytical path toward the construction of a glossary of green conservation encountered divergences and congruencies among the published meanings. Therefore, the process requires further investigation on the most complete and consistent versions toward a final form. After

the comparative analysis among equal terms, we propose a hierarchical classification to highlight the relationship of the terms according to their essential characteristics.

A map was created (Figure 7) to present the type of relationship that exists between the various lemmas and the position of the terms. Following the glossary structure, the visual classification incorporates the organization by main lemmas that still represent the basic element for more complex (sub)systems. The allocation of the terms required the evaluation of the quality of the definitions assigned to the same term; in this way, totally divergent meanings were flagged and discarded. Lastly, the correlation among the terms has been defined by the type of relationship. This resulted in a hierarchy among terms based on synonyms, inclusive terms (e.g., green includes "eco", "bio", "re"), and totally independent terms. Sustainability is the first core concept, from which comes the notion of green conservation in the last decades. Bio-oriented terms, not only for the number of constructs found in the literature but also for the ample declination of the lemma by the authors, are still the most challenging in linkages. If experts in the field or professionals were more aware of their conceptually different understanding, knowledge accumulation attempts may lead to less misleading results. We acknowledge that this way of coding the concept of green conservation by grouping the terms returns a partial image of the complex framework in which we live. Indeed, a great abundance of conceptualizations remains. However, we find it as a starting point for future deliberations.

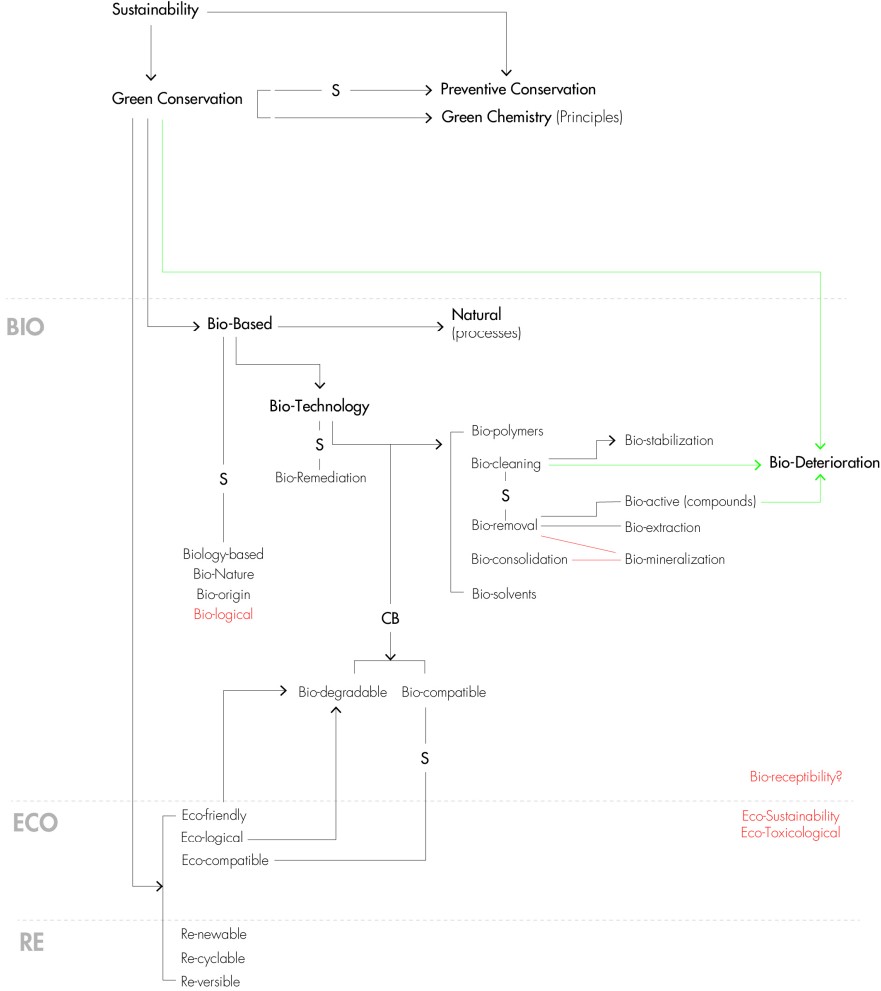

**Figure 7.** Hierarchical organization of terms according to the relation of (1) inclusion, (2) equality expressed as "S"—synonym, (3) potential derivation expressed as "CB"—Can be, and lastly, (4) transitivity, namely, when the term "acts for" a specific object. Terms in red are uncertain and demand further clarifications due to the lack of explicit definition.

### 4.3. Conclusions and Future Perspectives

Achieving more clarity is the biggest challenge of sustainability for cultural heritage; this paper seeks to overcome some of the vague conceptualizations that have clouded the efforts made to bring the green concept to the preservation field. Overall, we have gathered a comprehensive set of 128 definitions and 220 terms related to the concept of green conservation and systematically analyzed it against a coding framework to provide transparency regarding current green conservation understandings. While green building reviews have been published so far, no comprehensive and systematic analysis specifically on the cultural heritage field understandings was conducted prior to this study. We acknowledge that a green conservation understanding can be broader than the definition presented in our study if we consider other dimensions, such as the terminology from green marketing or other academic publications. However, the findings of this research can contribute to the coherence of the concept underlining commonalities and divergencies among a consistent part of experts in the field. The main goal of this work is to support the development of coherent strategies against the uncoherent use of the term, along with the introduction of novel products and services on the market. At present, the business sector is quickly evolving in trending concepts at the expense of knowledge. This is why a coherent notion of green conservation is of paramount importance for long-term survival. As previously argued, only a strong green conservation framework of understanding and practice can avoid dilution and improper use of the term. Behind the linguistic caviling and conceptual debates, this review identified six main knowledge syntheses of green conservation involving environmental, economic, and cultural values: (1) a process, (2) a change of culture, (3) preventive conservation, (4) a use of green materials, (5) an environmental analysis, (6) the revival of a tradition.

Our approach to systematically analyzing definitions provides the first quantitative evidence that green means many different things to different people. Worryingly, we found that only a third of definitions explicate a clear use and meaning of the term. As we confirmed, most of the scientific writings limit green conservation to the absence of toxicity for human health and the environment, minor but also consistent definitions should be more representative. Therefore, it is of great importance to understand the relationships among the terms and their semantic meanings. The next step will be to use a wider mapping to consolidate keywords and considerations across sustainability frameworks. In this view, good green implementation examples can help sharpen the understanding of the green conservation concept both among scholars and practitioners to shape informed green marketing in the cultural heritage sector.

**Author Contributions:** Conceptualization, A.T.; methodology, A.T.; investigation, A.T.; data curation, A.T.; writing—original draft preparation, A.T.; writing—review and editing, A.T.; visualization, A.T.; supervision, D.D.C. All authors have read and agreed to the published version of the manuscript.

**Funding:** This research received no external funding.

**Institutional Review Board Statement:** Not applicable.

**Informed Consent Statement:** Not applicable.

**Data Availability Statement:** The data presented in this study are available upon request from the corresponding author.

**Conflicts of Interest:** The authors declare no conflict of interest.

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
