# Peer review of "Towards a Reasoned Glossary of Green Conservation: A Semantic Review of Green-Oriented Terms in the Field of Cultural Heritage"

_sustainability, doi:10.3390/su151612104_

Round 1
Reviewer 1 Report
I am grateful for the opportunity to review this article. I congratulate the author on the topic, which bridges the gap between green practices and the conservation of cultural heritage.
In methodological terms the article seems to me to be well elaborated. The main issue is the engagement with theory. The manuscript intends to summarize the different definitions of the concept of green conservation to reconstruct a verifiable and comprehensive meaning of the term. This study can be seen as an attempt to answer the following questions: What do you understand by the expression “green conservation”? Is it clear or nebulous? Too vocal”? A “catch-all term”?
The fact that the article does not present a literature review chapter (the author followed the MDPI template) together with the absence of discussion and conclusions makes the 'article' more of a report than a scientific article. Thus, my recommendations are:
1. include a literature background chapter, however minimal and that minimally points to the methodology followed in the sense that one is testing/proofing something.
2. Include a discussion chapter where you establish an engagement with previous research, showing where and how you are contributing
3. Include a conclusions chapter which presents: theoretical contributions; practical implications; limitations and future research.
Author Response
Dear Reviewer,
We thank you for reviewing our work and for your precise comments. We appreciate you found this topic interesting and valuable for advancing research in the cultural heritage field. As engagement with theory seems to be the most crucial issue in your comments, we followed your suggestions to consolidate the structure of the work. You can find below our responses to your recommendations.
Point 1: The fact that the article does not present a literature review chapter (the author followed the MDPI template) together with the absence of discussion and conclusions makes the 'article' more of a report than a scientific article. Thus, my recommendations are: 1. include a literature background chapter, however minimal and that minimally points to the methodology followed in the sense that one is testing/proofing something.
Response 1: Since the theoretical background on the concept of green conservation was very short and briefly mentioned along with the introduction, we decided to split Section 1 into two parts. In this way, the novel sub-section, The development of a Green Sensibility in the field of Cultural Heritage (line 82), has its narrative autonomy to describe the evolution of the topic from the historical perspective. As stated in the first paragraph: “This section prefaces the semantic review with a brief overview of the most significant milestones that have led to the origin of the notion of green conservation. At the same time, it clarifies the state of the art to underline why a reasoned glossary is needed and why it differs from existing semantic reviews.”
Point 2: Include a discussion chapter where you establish an engagement with previous research, showing where and how you are contributing
Response 2: Given the absence of other literature reviews on the concept of green conservation, as far as we are aware, it is difficult to establish a discussion chapter on previous research. This is why the subsection mentioned above, also clarifies the state of the art on green glossaries where the term of green is conceived in a general sense.
Point 3: Include a conclusions chapter which presents: theoretical contributions; practical implications; limitations and future research.
Response 3: A conclusion chapter has been added, consequently to the subsection on limits of the work. The sub-section 4.3, Conclusions and future perspectives (line 664), resumes the overall source collection and presents the key findings of the research. Also, it suggests that future implementation can be found in the business sector where the concepts take on additional significance. This would build on the glossaries we have already done to confirm or generate novel semantic shades.
Reviewer 2 Report
The work with images would deserve more attention from the authors, the images could be better described in the text, e.g. in Figure 1, the authors could better explain the meaning of the color resolution and the size of the objects. Also, the choice of terms plotted on the y-axis is not clearly justified and defined.
Figure 2 has no telling power. Figure 3 or 8 is only an attractive form of visualization, etc.
Procedures and methods are not sufficiently justified in the contribution, for example, the authors have insufficiently clarified the form and principles of keyword selection (see e.g. figure 4), the selection of analyzed databases is not sufficiently justified either, the relationship between commercial databases and scientific databases is not sufficiently clear. Also, the previous research published in WoS, Scopus ProQuest, and other databases is not sufficiently analyzed. There is a high probability that the analysis of existing scientific publications could have a significant influence on the choice of key research terms. Although the authors provide a relatively comprehensive number of references, their citation, justification, and concrete contribution to the work is insufficient.
The problem that the authors are dealing with is current, but it is not a worldwide phenomenon, and the research would also benefit from a deeper regional and socio-social analysis.
I find the results shown in Figure 9 interesting and beneficial, unfortunately, they are based on a narrow set of insufficiently defined analyzed keywords. I recommend the authors expand the number of keywords and use cluster analysis in further research.
Author Response
Dear Reviewer,
We thank you for reviewing our work and for your precise comments. Also, recommendations for the figures visualization and their contents were much appreciated. Following your suggestions, more attention has been paid to the relationship between the text and the image, as well as rebalancing the contents of the different sections. More consistent references have been also added to consolidate the work. Please find below our responses to each point.
Point 1: The work with images would deserve more attention from the authors, the images could be better described in the text, e.g. in Figure 1, the authors could better explain the meaning of the color resolution and the size of the objects. Also, the choice of terms plotted on the y-axis is not clearly justified and defined.
Response 1: A clear explanation of the figure has been added in terms of colors and categorization. The size of the object refers to “the number of papers for each topic/session along the four editions of the conference 2015-2022”(line 239) and the color depends on the criteria of “a second subdivision was deemed necessary to differentiate biotechnologies (green) and nanotechnologies (blue)…” (line 241). To facilitate reading, a legend has been integrated into the figure.
Point 2: Figure 2 has no telling power. Figure 3 or 8 is only an attractive form of visualization, etc.
Response 2: Figure 2 was replaced with a new diagram, namely the road map of the methodology adopted for reviewing the scientific literature (line 294). We agree that the previous glossary overview was not exhausted. Therefore a new explanatory text (line 269) was added to describe the figure steps.
Figure 3 was removed as the number of contributions for each domain is clearly mentioned in the text.
Figure 8 was expanded to associate the definition of green conservation with the author.
Point 3: Procedures and methods are not sufficiently justified in the contribution, for example, the authors have insufficiently clarified the form and principles of keyword selection (see e.g. figure 4), the selection of analyzed databases is not sufficiently justified either, the relationship between commercial databases and scientific databases is not sufficiently clear. Also, the previous research published in WoS, Scopus ProQuest, and other databases is not sufficiently analyzed. There is a high probability that the analysis of existing scientific publications could have a significant influence on the choice of key research terms. Although the authors provide a relatively comprehensive number of references, their citation, justification, and concrete contribution to the work is insufficient.
The problem that the authors are dealing with is current, but it is not a worldwide phenomenon, and the research would also benefit from a deeper regional and socio-social analysis.
I find the results shown in Figure 9 interesting and beneficial, unfortunately, they are based on a narrow set of insufficiently defined analyzed keywords. I recommend the authors expand the number of keywords and use cluster analysis in further research.
Response 3: Changes in Section 2 have been made to better explain the procedures and methods of the work. As for the keyword selection, not only the term of green conservation has been detected but also other semantic units had to be involved in the process. Figure 4 is the synthesis of predominant “neighboring green terms” (line 263) found in the conference proceedings. The lemmas serve as a tool for grouping different words and allowing the evaluation of similar words.
Previous research on the semantic analysis of green conservation is lacking, as far as we are aware. Therefore it is difficult to review previous databases. We dedicate a short introductory paragraph to address that where we explain that “the language of green buildings still covers most of the existing vocabularies” (line 150) and most of the existing repositories such as “GreenSpec Directory or Oikos Green Building Source” (line 152) are not interesting for advancing the research in cultural heritage. They are a starting point as well as “Documented research exists on the semantic review of sustainability or circular economy terms (line 181) but unrelated to the CH field.
More references have been added to consolidate the meaning of green conservation. Moreover, “We acknowledge that a green conservation understanding can be broader than the definition presented in our study if we consider other dimensions such as the terminology from green marketing or other academic publications” (line 673). However, it is also true that we analyzed a consistent number of samples – 171 terms- from the only scientific domain which deals with the green conservation of cultural heritage. So, we agree that further research can certainly identify new implicit contexts and the business sector can offer a new cluster of terms.
Reviewer 3 Report
This article shows the results of a semantic survey and summarizes the different definitions of the concept of green conservation to reconstruct a verifiable and comprehensive meaning of the term.
One of the significant results of the article is a comparison of scientific and non-scientific bases, which allows a more complete assessment of social attitudes in relation to the issues studied by the authors.
However, specifics are lacking. Which articles were taken by the authors in these databases. Or at least the most striking examples can be welcome to confirm your data. The link to the database is not equivalent to the number of sources studied, stated in the annotation. This issue needs to be corrected.
For example, "Figure 1. Topics comparison over the four editions of the Green Conservation Conference 2015-2022. The table reports the number of papers for each topic, and it combines overlapping sessions under three main categories: new bio-based materials and biotechnology, nanotechnologies, and sustainable practices." What kind of articles do you mean?
Page 11, line 395: reference to (Figure 10) but there are nine figures.
Figure 7 is a Table in structure.
The authors can refine the text in terms of language, as some sentences can be written more clearly.
for example line 30-32. "However, the term green, before attaining the contents of the modern concept, was limited in its most literal sense to describing a specific disciplinary field, that of the environment and natural systems." rearrange the words "the term green". Get "However, before attaining the contents of the modern concept, the term green was limited in its most literal sense to describing a specific disciplinary field, that of the environment and natural systems."
Line 35 : "For this reason it should be stresses that when...". You need to put a comma and change the form of the verb: "For this reason, it should be stressed that.."
Line 39 "A first historical" - "The first historical..."
These are examples from the beginning of the text, further along the text errors occur regularly. The author needs to improve the text.
Author Response
Dear Reviewer,
We thank you for reviewing our work and for your precise comments. Also, recommendations for the figures visualization and their contents were much appreciated. Following your suggestions, more attention has been paid to the references – that were implemented - and to the relationship between the text and the image. Also, a rebalancing of the contents of the different sections was done to validate the importance of both databases. A second English review was also carried out to provide more clear sentences. Please find below our responses to each point.
Point 1: Which articles were taken by the authors in these databases. Or at least the most striking examples can be welcome to confirm your data. The link to the database is not equivalent to the number of sources studied, stated in the annotation. This issue needs to be corrected. For example, "Figure 1. Topics comparison over the four editions of the Green Conservation Conference 2015-2022. The table reports the number of papers for each topic, and it combines overlapping sessions under three main categories: new bio-based materials and biotechnology, nanotechnologies, and sustainable practices." What kind of articles do you mean?
Page 11, line 395: reference to (Figure 10) but there are nine figures.
Figure 7 is a Table in structure.
Response 1: Changes have been made to Figure 1 to better explain the number and type of papers selected for the database. A total of 126 papers was selected and one of the first actions of the analysis consisted of their clusterization according to the topic of the research. In short, the methodology tracks the following actions:
“1. the identification of the research context among the topics of the conference to chart the evolution of green conservation storytelling over the years;
- the definition of three main categories for recurrent topics to merge the papers under the same session and avoid overlapping;
- the construction of the glossary through a series of different entries that consider (a) the lemma to which the term belongs, (b) the conference year, (c) the bibliographic reference or authors, (d) the scientific field among the three previous categories, (e) the term and, where applicable, the object of the construct, and lastly, (f) the as found definition;
- the completion of the semantic collection, with an accent on the difference between explicit and implicit descriptions for each term;
- the identification of definition synthesis." (line 271)
Second, further adjustments have been made to Figure 7 and to the reference of Figure 10, as highlighted.
Round 2
Reviewer 1 Report
Congrats on the revision. Well done.
Author Response
Dear Reviewer,
Thanks for accepting the revised work and for your positive remarks on the manuscript.
We really appreciate the maximum rating of the contents and methods of the article during the evaluation process. Also, we hope our research can guide future works.
Reviewer 2 Report
I thank the authors for the quality of the required editing of the post. The authors responded factually to the reviewer's comments. They added the required information, increased clarity, and explained the procedures more broadly. The authors have sufficiently expanded the cited sources and relevant literature. I recommend the article for publication.
Nevertheless, I recommend working more with research that is published in scientific databases in the future.
Author Response
Dear Reviewer,
Thanks for your positive remarks on the quality of our revisions and for recommending the work for publication. Your previous comments helped us to define a more clear narration and to support the semantic review with a coherent number of references.
We considered all the suggested changes and, certainly, this work will not be enclosed in itself. Rather, it will guide the development of our current research on the topic of sustainable and greener cultural heritage.
Reviewer 3 Report
The author has significantly improved the article, clarified his own position. The study has become readable, there are no omissions in the methodology In the previous version, we had to figure out what the author means. The author needs to make minor adjustments: The introduction should be restructured. The purpose and description of the structure of the article (Section) should be at the end of the Introduction. Otherwise, logic is violated. Relevance and state of affairs, the structure of the article, after - relevance and state of affairs. Line 39 "This leads..." - a noun should be added.Author Response
Dear Reviewer,
Thanks for your positive remarks on the improvements and for appreciating our effort in a better illustration. Your previous suggestions guided the work toward a more clear narration in terms of methods, images and references.
Please find below the response to the minor editing you required. We hope the current structure of the article follows the correct logic.
Point 1: The author needs to make minor adjustments: The introduction should be restructured. The purpose and description of the structure of the article (Section) should be at the end of the Introduction. Otherwise, logic is violated. Relevance and state of affairs, the structure of the article, after relevance and state of affairs.
Response 1: We agree on the reorganization of the introduction. For this reason, we moved the description of the sections to the end of it (see line 178). In this way, the introductory part can logically conclude with a preview of the methods and results of the article.
Point 2: Line 39 "This leads..." - a noun should be added.
Response 2: Thanks for the note. We completed the sentence in this way: (line 39) “This semantic blurriness leads to imprecise definitions…”